# The Management of Children with Chronic Health Problems at School from the Perspective of Parents

**DOI:** 10.3390/nursrep15020057

**Published:** 2025-02-06

**Authors:** María del Pilar Morales Ruíz, Isabel María Fernández-Medina, Antonio Javier Alias-Castillo, María Isabel Ventura-Miranda, María del Mar Jiménez-Lasserrotte, María Dolores Ruíz-Fernández

**Affiliations:** 1Melilla Regional Hospital, 52005 Melilla, Spain; mmr582@inlumine.ual.es; 2Department of Nursing, Physiotherapy and Medicine, Faculty of Health Sciences, University of Almería, 04120 Almería, Spain; aac142@inlumine.ual.es (A.J.A.-C.); mvm737@ual.es (M.I.V.-M.); mjl095@ual.es (M.d.M.J.-L.); mrf757@ual.es (M.D.R.-F.); 3Facultad de Ciencias de la Salud, Universidad Autónoma de Chile, Temuco 4780000, Chile

**Keywords:** children, chronic health problems, school nurses, parents’ experiences, qualitative study

## Abstract

**Background:** The prevalence of chronic health problems in childhood, such as food and respiratory allergies, as well as endocrine and skin disorders, has increased globally. Parents and children experience difficulties in managing their illness that reduce their quality of life and increase the presence of negative feelings such as fear or anxiety during school hours. **Objectives**: The aim of this study was to describe and understand the parents’ experiences of the management of children with chronic health problems during school hours. **Methods**: A qualitative descriptive study design was used. Semi-structured interviews were conducted with 17 parents (1 male and 16 female) who have children with a chronic health problem between September 2022 and June 2023. The data were analyzed with the help of a qualitative data analysis software. The COREQ checklist was used to develop the study. **Results**: Two categories and four subcategories emerged from the data analysis: (1) Parents’ Perceptions, with the subcategories 1.1., The School Nurse, and 1.2., Educational Environment; and (2) Chronic Pediatric Health Problems, with the subcategories 2.1., Emotional Impact, and 2.2., Lack of Resources. **Conclusions**: Children with chronic health problems are unprotected during school hours due to the absence of a school nurse. Parents and teachers have to cope with their care without the necessary knowledge, which makes it difficult to manage children with chronic health problems in the school environment.

## 1. Introduction

In recent decades, there has been an increase in the prevalence of chronic health problems in the pediatric population. It is estimated that approximately 25% of children have a chronic health problem [1]. Chronic health problems in children are those that last longer than three months, have an impact on normal activity and require medication, a special diet, medical technology, assistive devices or personal help to compensate, leading to a greater need for the use of health services than most children of a similar age [2]. Chronic health problems are broadly characterized by their prolonged duration and their lack of a spontaneous and complete cure [3]. The most frequent chronic health problems in childhood are type 1 diabetes, food and respiratory allergies, asthma, epilepsy, celiac disease, cancer and cerebral palsy [4]. Of the chronic health problems mentioned above, type 1 diabetes and food allergies are the most common. About 6–8% of the child population is affected by a food allergy [5], and 25 out of 100,000 children have type 1 diabetes worldwide [6]. These pathologies are not only important because of their high incidence but also because of their consequences on children’s health, such as anaphylaxis in food allergies and hypoglycemia or ketoacidosis in type 1 diabetes. To prevent complications such as those mentioned above, constant care and proper management of the problem are necessary. Type 1 diabetes requires constant monitoring of glucose levels, a specific diet and insulin administration. A correct dose of insulin depends on the level of glucose, the dose of carbohydrates consumed, planned physical activity and the general condition of the child. Insulin therapy reduces the risk of acute metabolic disorders and must be administered appropriately [7]. In the case of food allergies or celiac disease, care includes dietary modification, knowledge of food preparation to avoid cross-contamination and an individualized emergency plan that includes using an epinephrine auto-injector in case of an allergic reaction [7,8].

Although the impact of the chronic health problem on children and parents depends on the severity and the nature of the problem, it produces a severe alteration of a family’s daily routines that affects health-related quality of life [9]. For families of children diagnosed with a chronic health problem, an important part of the process of adjusting to the disease is integrating its management into daily routines, including school [10]. The effective management of chronic health problems represents a daily challenge for parents, many of whom suffer from stress, anxiety and exhaustion [11]. Approximately half of a child’s waking hours are spent at school. During this time, children with chronic health problems must not only attend compulsory classes but also self-manage their illness by closely monitoring their blood glucose levels or following dietary recommendations [12]. Therefore, close collaboration between parents, teachers, school nurses and school administration is necessary for the proper management of chronic health problems in the school environment for the normalization and integration of both the child and the chronic health problem [13].

School nursing is defined as a specialized field of public health nursing that protects and promotes the health of students, enabling their normal development and promoting academic success between the ages of 5 and 19 [14]. Therefore, the school nurse is a leader in the development, implementation and evaluation of school health programmes, including the management of student health problems. In addition to avoiding the repercussions of a chronic health problem, the presence of nurses in schools can reduce the common consequences that affect all students with chronic health problems, such as bullying, absenteeism or poor academic performance [15]. School nurses also play a key role in empowering children, families and teachers to manage the condition by providing them with the necessary information, resources and support. However, not all schools currently have a school nurse [16]. Although community nurses visit schools when there is no formal school nurse position in a school [17], the American Academy of Paediatrics advocates for a minimum of one full-time school nurse in each school [18]. Moreover, parents of school-age children recognize the benefits of having a school nurse in their schools [19]. The aim of this study is to describe and understand the parents’ experiences on the management of chronically ill children during school hours. The research was conducted qualitatively because we wanted to know the opinions and perspectives of parents with children with chronic health problems and a quantitative study did not allow us to explore this type of concept [7,8].

## 2. Materials and Methods

### 2.1. Study Design

A descriptive study was carried out to explore the experiences and opinions of parents. This methodology permits the understanding of a phenomenon in a natural environment, insisting on exploring the meaning of the people involved. The study was developed according to the Consolidated Criteria for Reporting Qualitative Research (COREQ) [20].

### 2.2. Participants and Setting

Parents from an area of southeastern Spain were recruited through purposeful sampling. Information about the study was disseminated to different primary educational centers. Parents interested in participating in the study contacted the first author via phone to schedule an in-person meeting. The inclusion criteria were as follows: being a parent with a child between 6 and 12 years old attending primary school and suffering from a chronic health problem such as diabetes, allergies, asthma or any other condition. Parents who did not meet the above criteria or those who refused to participate in the study were excluded.

### 2.3. Data Collection

Data collection was completed between September 2022 and June 2023 through individual semi-structured interviews. The interviews were conducted face-to-face by the first author of the study in a quiet, private room attached to the school premises. To conduct the interviews, an interview guide with open-ended questions based on a literature review and the authors’ previous experience in childcare was used. The questions were related to different aspects of school-based management of pediatric chronic health problems (Table 1).

The interviews were initiated by obtaining sociodemographic data from the participants, followed by a general question about their experiences and opinions regarding the management of their child’s chronic health problem during school hours. Each participant was only interviewed once, and parents who were a couple were interviewed separately. The duration of the interviews was approximately 30 min. The interviews were audio-recorded and transcribed verbatim. Data collection ceased when data saturation was reached in Interview 15. However, two more interviews were conducted for verification.

### 2.4. Data Analysis

The computer-assisted qualitative data analysis programme Atlas.Ti Version 9.0 was used for data management and analysis. Firstly, significant fragments were selected as quotations, organized and coded according to their information by the researchers. Multiple memos were written to establish characteristic data segments and their relationships during the coding process, allowing the researchers to document their thoughts, interpretations and reflections on the data collected during the research process. In the interpretive phase of the analysis, researchers established patterns of similarities and differences within the data [21]. Significant fragments of data were grouped into categories and themes according to their similarities. Data analysis was carried out by two researchers independently and similarities and differences were discussed until agreement was reached. The qualitative data analysis was used to facilitate the comparison and understanding of meaningful fragments. A table with semantic units, subcategories and categories was elaborated during this stage (Table 2). Finally, a third researcher verified the analysis.

### 2.5. Rigour

The rigour of this study was confirmed through the criteria of confirmability, credibility, reliability and transferability [22]. The interviews were conducted by a researcher who did not know the participants, and confirmability was corroborated. To corroborate credibility, the interviewees obtained a copy of their transcript to ratify their answers. The interpretation of the data was based on the triangulation of the participants, and the analytical process was verified by two independent reviewers who reviewed the conclusions, discussing similarities and differences, thus confirming reliability. Regarding transferability, a thorough description of the study setting, participants and method was made.

Participants were informed of the aim of the study and signed an informed consent form. The confidentiality, anonymity and privacy of the participants were guaranteed.

## 3. Results

In total, 17 parents (16 mothers and 1 father) participated in the study. None of the participants dropped out of the study. The mean age of the participants was 45 years-old. A total of 53% of the participants have children with diabetes, 5% have children with food intolerances, 29% have children with food and respiratory allergies and the rest have children with other chronic diseases such as psoriasis, phenylketonuria and hypothyroidism. The participants’ sociodemographic characteristics are shown in Table 3. From the data analysis, three themes and six sub themes emerged. All of them helped us describe and understand parents’ experiences and opinions about managing children with chronic health problems during school hours.

### 3.1. Parents’ Perceptions

All parents had a positive attitude towards having a nurse in the school. They thought that having a nurse is very important to help their child to manage the chronic health problem as the health care of children with chronic pathologies during the school day depends on the knowledge and actions of teachers, whose training is based on teaching and not on health care. This theme shows the attitudes, opinions and perceptions parents with children with chronic health problems have about the school nurse.

#### 3.1.1. The School Nurse

According to the parents who participated in the study, the care of their children is reduced while they attend school. They feel their children are unprotected during school hours. As a result, parents stated that a school nurse is essential for the care of children with chronic health problems. There are more and more children with chronic disorders, requiring a protective measure such as school nursing, which detects, prevents and can solve health problems of the students themselves. They agreed that if a nurse were present during school hours, the quality of life of their children would improve and parents and teachers would be calmer.

*“If there is a school nurse the children will feel safer, they know that they are sick and that their parents are not there. It’s not the same for them to feel alone as it is to feel that there’s a nurse who knows about their illness. It would also be very reassuring for the parents”*.(Participant 6)

The participants told us that integrating nursing into schools would be an effective and efficient added value, who, together with the teaching staff, would be able to manage the daily life of the schoolchildren. They argue that school nurses have the ability to integrate patient-centred knowledge and skills and communicate with students, teaching staff and community resources, not to mention their ability to think critically. Parents were sure that a school nurse would be familiar with the chronic health problems and the most characteristic symptoms that their children may suffer from, how they usually evolve, what medication they may need and how to administer it, even in severe cases such as severe hypoglycaemia or anaphylactic shock, and could accompany the child in acute moments of their pathology.

*“Well, apart from controlling his illness and knowing how to treat it, I think a school nurse knows more than us and can even teach my son’s classmates the process of his illness. In some crises the child can be playing in the school playground or in the street with his or her friends, and they would know how to treat him if a nurse has taught them”*.(Participant 2)

#### 3.1.2. Educational Environment

Parents in our study expressed agreement with the teachers’ attitude towards their children’s health conditions but acknowledged that, although they care deeply about their students, they are not health professionals and are limited in their knowledge and resources when faced with situations related to their children’s chronic health problems.

*“They are interested, the teacher is great, but she can’t do more because she can’t take responsibility for my daughter’s illness as she neither has the qualifications nor the knowledge”*.(Participant 6)

The interviewees reported that when there is a problem related to their children’s health, the teachers call the parents to resolve the situation, but many of them are unable to pick up the phone, and not all of them are able to immediately leave work and go to the school in order to resolve the situation. Other parents said they have had to stop working to attend to their children’s complications. In view of the above, the participants stressed that the incorporation of a nurse in a school would mean an important change in the lives of families, especially as it would prevent absenteeism of children and work complications for parents.

*“The teacher would call me and tell me that she had a high or low blood sugar, I had to go to the school every day. I work cleaning houses, and I had to go and give her insulin. I’ve been out of work because I’ve had to take a lot of time off because of my daughter’s illness. I have had a very hard time as I have no help from anyone”*.(Participant 11)

### 3.2. Chronic Pediatric Health Problems

Parents of children with health problems experience many negative feelings, with anxiety, distress, apprehension of the unknown, restlessness, fear, insecurity and even social prejudice being the most prevalent. Students with chronic health problems also tend to have more emotional problems than children without such problems. This theme shows parents’ perceptions and experiences of the emotional aspects surrounding children with chronic health problems and their families, especially during school hours.

#### 3.2.1. Emotional Impact

Our participants showed that their children often feel rejected at school, see themselves as different from their peers and feel uncomfortable wearing a blood glucose monitor and therefore want to hide it. As a result, the side effects of their own illness cause them physical and psychological discomfort. Participants highlighted the fact that a school nurse is uniquely placed to work on dimensions related to self-esteem, social acceptance and mood in children with chronic health problems.

*“My daughter feels bad because she tells me that the children don’t know what diabetes is and why she has to take insulin… if there was a nurse in the school, she could also help her to do this”*.(Participant 2)

The participants in our study felt somewhat helpless in terms of the management of their children’s health problems by the educational system. The feeling of abandonment was expressed by all the interviewees. They also told us that they sometimes feel anguish, fear and panic in case their children have a problem at school related to their health problem and the teachers are not able to resolve the situation quickly. According to the point of view of our participants, families feel helpless in terms of their children’s health while they are at school.

*“When I hear the ambulances, I think of my daughter because she is not safe as she doesn’t have a nurse, she doesn’t have anyone, the teachers don’t understand any of that because they have never seen any of those things, so I worry because she is not safe, it’s a school that doesn’t have a nurse, that’s what it is”*.(Participant 14)

#### 3.2.2. Lack of Resources

Most of the participants in our study reported great satisfaction with the support received from the school; however, they highlighted the considerable lack of resources when schools have to address the needs of their children’s chronic health problems. They therefore expressed their desire to invest financially in improving the conditions and resources of the educational centers in order to improve the quality of life of their children.

*“In Spain there is a great lack of resources in schools, not only at the educational level, but also at the health level, most of them lack not only nurses but also resources in an emergency situation. With greater investment, school nurses could be hired and the centers could be provided with the means to attend children during school hours when necessary”*.(Participant 8)

In addition, the participants mentioned that they lack a competent health professional such as a school nurse to ensure that their children’s needs are met. With a nurse, teachers would not have to take on responsibilities that do not belong to them, and the children would be able to take care of themselves.

*“The other day at 8 a.m. I had to change my daughter’s glucose sensor because it suddenly stopped working. I took her to school with a newly installed sensor, and you feel uneasy. It is true that as we have so much confidence with the teacher, I write to her at any time and say ‘Please, give her something if you notice that she has a low (glucose) level, and she answers me immediately… But, if there were a nurse at the school, I wouldn’t care if the sensor was broken because I would know that she would check it later”*.(Participant 2)

## 4. Discussion

The aim of this study was to describe and understand parents’ experiences of the management of children with chronic health problems during school hours. The participants in the study reported that they felt their children were unprotected during school hours, especially due to the absence of a nurse. The study by Mustafa et al. [23] showed that one in four parents perceive that their children are not safe in the school environment. According to the results of other studies, the presence of a nurse in schools reduces stress and anxiety for parents of children with chronic health problems [13]. Therefore, the lack of school nurses is thought to increase the psychological burden experienced by the parents of these children.

In the case of chronically ill children, collaboration between parents and teachers is necessary to ensure the child’s safety. Thus, it is essential that schools engage with both students and parents to assess the risk of chronic health problems [12]. During school hours, parents have to entrust their children’s well-being to teachers. However, the parents in our study told us that teachers do not have the knowledge to manage their child’s chronic health problem or to deal with an emergency situation. This could be because many teachers do not understand the importance of chronic health problems and are indifferent to the information provided [7]. The study carried out by Herbert et al. [24] revealed that less than a quarter of parents thought that teachers were able to meet the needs of their children affected by type 1 diabetes mellitus and only half of parents believed that school staff could manage their child’s chronic health problem. As in our study, other evidence has highlighted parents’ lack of trust in teachers in the proper handling of an emergency [25]. Studies with teachers have found that they do not have the necessary knowledge and are not prepared to intervene in emergencies such as the administration of intramuscular medication in situations of anaphylaxis [26]. On the other hand, in the study of Muñoz (2018), some teachers state that it is not their responsibility to check or control symptoms of chronic health problems or to administer epinephrine in emergencies [27]. As noted in other studies, the lack of knowledge is a barrier to chronic disease management [28].

As in other studies, our participants emphasized that teachers do not have sufficient experience to meet the needs of their children, underestimating the risk of complications, which represents a potential threat to the lives of children with chronic health problems [29]. The lack of competence of teachers regarding the needs of children with chronic health problems combined with the absence of nurses in schools means that families have to come to the school to administer treatment to their children or in the case of an emergency [30]. Parents have significant fears about potential emergencies occurring when their children are at school. Parents also perceive a lack of resources and equipment, which increases their fear and insecurity [24].

As found in this study, parents are directly responsible for the care of their child during school hours, sometimes having to be absent from work. Parents therefore have to be available at all times for any event occurring in the school environment [29,31]. This results, as in our study, in some parents having to give up work in order to be constantly contactable and to be able to meet their children’s needs and ensure their safety [32]. As a result, parents are calling for the inclusion of a nurse in schools. In line with the results of other studies, participants think that the presence of a school nurse would improve their children’s ability to manage their illness and adjustment to school [24,33]. In particular, the school nurse would play a key role in decreasing respiratory symptoms, allergic reactions, health-related absenteeism and increasing medication knowledge and the availability of student-specific injectable devices [34]. However, the presence of a school nurse in the Spanish context is very rare [35], and parents are therefore demanding the development of policies that facilitate the integration of school nurses to facilitate health care for students with chronic health problems [36].

### 4.1. Limitations

This study is not without limitations. Firstly, it only reflects the experiences and opinions of parents in southern Spain, and parents’ experiences and opinions in other geographical areas or countries could differ. Future research should include the experiences and opinions of teachers and school nurses as well as parents from other geographical areas. Secondly, most of the participants were women, so their perspective could influence the results. This was because most of the parents who were invited to participate in the study declined to talk about it. Future studies should include the fathers’ perspectives to avoid the influence of gendered perspective in the interviews. Thirdly, the sociodemographic characteristics of the sample are homogeneous, which could have limited the variability of responses. Finally, in two of the interviews conducted, the lack of verbal fluency and command of Spanish made data collection difficult.

### 4.2. Scientific and Professional Contributions

This study provides insights into the experiences that parents face when managing children with chronic health problems in schools. The study highlights the need for a school nurse for the appropriate management of children with chronic health problems. In addition, the results of this study provide school nurses with information and better knowledge about the resources needed by children with chronic illnesses and the experiences of their parents, which is key to improving health outcomes. It may be necessary to provide training for teachers or improve existing training to make children and parents feel safer at school.

## 5. Conclusions

Children with chronic illnesses need constant attention and vigilance during school hours. However, parents feel that their children are unprotected due to the absence of a school nurse. The educational system is characterized by a lack of resources and teacher preparation. In consequence, parents and teachers have to cope with their care without the necessary knowledge, which makes it difficult to manage children with chronic health problems in the school environment. All of this has a great emotional impact not only on the children but also on the parents, with some mothers abandoning their jobs to be available to care for their children during school hours. This situation causes families to demand the presence of a school nurse who can carry out preventive and care activities, allowing educational–health integration and improving the quality of life of children with chronic diseases and their families.

## Figures and Tables

**Table 1 nursrep-15-00057-t001:** Interview protocol.

Stage of the Interview	Subject	Examples of Questions
Introduction	My intention	I am a member of a research group, which study the experiences, attitudes and opinions of parents with chronically ill children. Knowing their experiences and opinions could be useful for improving the care of chronically ill children during school hours.
	Information and ethical issues	Your participation is voluntary. We need to record the conversation in order to analyze the data. Your personal data will not be disclosed.
	Consent	Signing of the informed consent.
Beginning	Introductory question	Tell me about your experience as a parent of a chronically ill child in terms of the care your child has received at school.
Development	Conversation guide	How do you think the presence of a nurse in the school could help? How has the relationship with the teachers been in terms of the treatment and management of your child’s illness?What serious situations has your child experienced during school hours regarding complications of his/her own illness?How do you experience the fact that your child is in school hours without a health care provider?How would you describe the current situation in schools for children with a chronic illness?
Closing	Final question	Would you like to add something else?
	Appreciation	Thank you for your participation. You will receive a copy of the study once it has been completed.

**Table 2 nursrep-15-00057-t002:** Categories, subcategories and semantic units.

Category	Subcategory	Semantic Unit
Parents’ perceptions	School nurse	Lack of protection, need for school nurse, reduced quality of life, nurse–teacher collaboration, roles of the school nurse, urgent situations.
	Educational environment	Teachers’ lack of knowledge, children’s health problems, parental presence in school.
Chronic pediatric illnesses	Emotional impact	Children’s feelings of discomfort and rejection, social acceptance, parents’ fear, feelings of abandonment and helplessness.
	Lack of resources	Satisfaction with school support, lack of resources, financial investment.

**Table 3 nursrep-15-00057-t003:** Participants’ sociodemographic characteristics.

Gender	
**Male**	1
Female	16
**Age**	
30–39	6
40–49	10
>50	1
**Civil status**	
Married	12
Single	2
Divorced	3
**Number of children**	
1	1
2	8
3	6
>3	2
**Chronic disease**	
Diabetes mellitus	9
Food intolerances	1
Food and respiratory allergies	2
Others	5
**Years with chronic health problem**	
<1	5
1–2	4
2–3	2
>3	6

## Data Availability

The raw data supporting the conclusions of this article will be made available by the authors on request.

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
