# Peer review of "The Management of Children with Chronic Health Problems at School from the Perspective of Parents"

_nursrep, 2025, doi:10.3390/nursrep15020057_

Round 1
Reviewer 1 Report
Comments and Suggestions for Authors
I read the manuscript with great interest.
Please revise the literature.
Please let us know if you observe any differences in perceptions based on the number of years the child has been chronically ill.
Author Response
I read the manuscript with great interest.
Thank you for your comments.
Please revise the literature.
Thank you for your suggestions. The literature has been revised.
Please let us know if you observe any differences in perceptions based on the number of years the child has been chronically ill.
Thank you for your comments. We have appreciated that parents of chronically ill children with fewer years of schooling were more worried and fearful about the problems that could arise at school.
Reviewer 2 Report
Comments and Suggestions for Authors
The study has important methodological limitations that need to be addressed to be published:
Firstly, the study population is characterised by absolute homogeneity, which precludes the definition of a study group as 'fathers' in the absence of mothers. The researchers are required to provide a justification for the non-inclusion of male parents in the study, and to take into account the influence of gender in the study population. The study population should be expanded to include men, or the population should be redefined in such a way as to include gender as a study variable.
Secondly, there are no triangulation mechanisms in place, which must be implemented and explained.
Furthermore, the absence of double-checking mechanisms is notable, and it is imperative that these are implemented and explained.
Additionally, the analysis of the memos must be accompanied by a detailed explanation.
In addition, the conceptual map of the categories should be incorporated to elucidate their interrelationships.
Author Response
Firstly, the study population is characterised by absolute homogeneity, which precludes the definition of a study group as 'fathers' in the absence of mothers. The researchers are required to provide a justification for the non-inclusion of male parents in the study, and to take into account the influence of gender in the study population. The study population should be expanded to include men, or the population should be redefined in such a way as to include gender as a study variable.
Thank you for your comments and suggestions. The inclusion of few men has been added as a limitation of the study. Future research should include more experiences of men.
Secondly, there are no triangulation mechanisms in place, which must be implemented and explained.
Furthermore, the absence of double-checking mechanisms is notable, and it is imperative that these are implemented and explained.
Thank you for your comments. The double-cheking mechanisms have been included in the data analysis.
Additionally, the analysis of the memos must be accompanied by a detailed explanation.
Thank you for your suggestions. We have added more explanation about memos.
In addition, the conceptual map of the categories should be incorporated to elucidate their interrelationships.
Thank you for your comments and suggestions. A table with themes, categories and units of meaning has been incorporated.
Reviewer 3 Report
Comments and Suggestions for Authors
Dear Authors,
Thank you for the opportunity to review your research. The research addresses a very useful and important topic that is increasingly relevant today. It is also significant that the research is based on qualitative research to capture the opinions of parents who spend the most time with their child and deal with the child's health condition.
Introduction is clearly structured and well theoretically explained. The methodology is elaborated in accordance with the rules of qualitative research. The obtained results are clear and well-presented, as is the discussion. In the discussion section, the authors clearly link the obtained results with recent research and provide a comprehensive conclusion to their study.
Recommendation:
Add a section on the limitations of the study and the scientific and professional contributions.
It seems to me that the references in the text do not numerically correspond to the references in the bibliography; for example, line 85/86 "The study was developed according to the Consolidated Criteria for Reporting Qualitative Research (COREQ) [20]. Reference 20 is Rodriguez, E., Rivera, D.A., Perlroth, D., Becker, E., Wang, N.E., Landau, M. School nurses’ role in asthma management, school absenteeism, and cost savings: a demonstration project. J Sch Health 2013, 83(12), 842-850.
I believe it should be reference 21 - Tong, A.; Sainsbury, P.; Craig, J. Consolidated criteria for reporting qualitative research (COREQ): a 32-item checklist for interviews and focus groups. Int J Qual Health Care 2007, 19(6), 349-357.
I think the references are shifted by one number. Please check the references.
Author Response
Thank you for the opportunity to review your research. The research addresses a very useful and important topic that is increasingly relevant today. It is also significant that the research is based on qualitative research to capture the opinions of parents who spend the most time with their child and deal with the child's health condition.
Introduction is clearly structured and well theoretically explained. The methodology is elaborated in accordance with the rules of qualitative research. The obtained results are clear and well-presented, as is the discussion. In the discussion section, the authors clearly link the obtained results with recent research and provide a comprehensive conclusion to their study.
Recommendation:
Add a section on the limitations of the study and the scientific and professional contributions.
Thank you for your comments and suggestions. We have added a section on the limitations of the study and the scientific and professional contributions.
It seems to me that the references in the text do not numerically correspond to the references in the bibliography; for example, line 85/86 "The study was developed according to the Consolidated Criteria for Reporting Qualitative Research (COREQ) [20]. Reference 20 is Rodriguez, E., Rivera, D.A., Perlroth, D., Becker, E., Wang, N.E., Landau, M. School nurses’ role in asthma management, school absenteeism, and cost savings: a demonstration project. J Sch Health 2013, 83(12), 842-850.
I believe it should be reference 21 - Tong, A.; Sainsbury, P.; Craig, J. Consolidated criteria for reporting qualitative research (COREQ): a 32-item checklist for interviews and focus groups. Int J Qual Health Care 2007, 19(6), 349-357.
I think the references are shifted by one number. Please check the references.
Thank you for your comments and suggestions. We have checked the references.
Reviewer 4 Report
Comments and Suggestions for Authors
1. References at the end of the work do not begin with 1.
2. In table 2, given segment 3 (3.1 and 3.2) indicate what are 1 and 2
3. If it is possible to emphasize in a few sentences the difficulties of teachers in working with these children, but from the teachers' point of view (their views) - through the available literature
4. in such studies, where decision-makers should see the shortcomings, it is important to refer to WHO and/or EU documents
Author Response
1.References at the end of the work do not begin with 1.
Thank you for your comments. Sorry, it is a mistake. The number of the references has been modified.
2. In table 2, given segment 3 (3.1 and 3.2) indicate what are 1 and 2
Thank you for your suggestions. The numbers of the table 2 have been modified.
3. If it is possible to emphasize in a few sentences the difficulties of teachers in working with these children, but from the teachers' point of view (their views) - through the available literature
Thank you for your comments, the article has been made taking into account the views of parents, in the discussion section we have included literature concerning teachers' difficulties from their point of view. However, a study of teachers' views on the subject would be necessary.
4. in such studies, where decision-makers should see the shortcomings, it is important to refer to WHO and/or EU documents
Thank you for your comments and suggestions. We have made reference to Council on School Health, Australian Nursing and Midwifery Federation and National Association of School Nurses because we have not found WHO or EU documents referred to the subjet.
Reviewer 5 Report
Comments and Suggestions for Authors
Hi, Dear Editor,
Thank you for giving me the opportunity to review the article entitled " The management of children with chronic health problems at school. A qualitative study”.
It is a good and important topic, but it needs some corrections, which I will discuss below.
It would be better to write in the title: From the perspective of parents of children with chronic health problems.
Researchers need to provide a clear reason why the research was conducted qualitatively when it could have been done quantitatively.
The background should explain the problem and issue more clearly.
The questions state that the parents' opinions and experiences were examined. It must be determined whether the experience was examined or the opinions, because after that the study approach will change.
Theme and categories were not well-formed.
The theme should be abstract and the items mentioned are not abstract.
Also the sub-theme, which should be named a category, and the items mentioned are not categories.
In general, Table 2 needs serious revision. And consequently, the discussion will also undergo changes
Instead of a unit of meaning, it should be written as meaning unit
Author Response
Hi, Dear Editor,
Thank you for giving me the opportunity to review the article entitled " The management of children with chronic health problems at school. A qualitative study”.
It is a good and important topic, but it needs some corrections, which I will discuss below.
It would be better to write in the title: From the perspective of parents of children with chronic health problems.
Thank you for your suggestions. The title of the paper has been modified.
Researchers need to provide a clear reason why the research was conducted qualitatively when it could have been done quantitatively.
Thank you for your comments and suggestions. The research has been conducted qualitatively because we wanted to know the opinions and perspectives of the parents with children with chronic health problems and a quantitative study did not allow us explore this type of concepts.
The background should explain the problem and issue more clearly.
We appreciate your suggestions. The background section has been revised.
The questions state that the parents' opinions and experiences were examined. It must be determined whether the experience was examined or the opinions, because after that the study approach will change.
Thank you for your suggestions. The aim of the study has been revised and we have explored the experiences of the parents.
Theme and categories were not well-formed.
The theme should be abstract and the items mentioned are not abstract.
Also the sub-theme, which should be named a category, and the items mentioned are not categories.
Thank you for your considerations, we have changed the denomination, however, we have intend to give a name to the themes and categories which represent the text and the experiences of participants.
In general, Table 2 needs serious revision. And consequently, the discussion will also undergo changes
Thank you for your suggestions. The table 2 has been revised.
Instead of a unit of meaning, it should be written as meaning unit
Thank you for your comment. The term has been changed.
Round 2
Reviewer 5 Report
Comments and Suggestions for Authors
Hi dear editor
Thankful
The authors' response to the article entitled “The management of children with chronic health problems at school. A qualitative study “was carefully reviewed. The corrections were made correctly, but there are still two issues that concern me and it needed to check again by authors, so, I will state below.
1. Researchers need to provide a clear reason why the research was conducted qualitatively when it could have been done quantitatively.
The authors have responded in the letter, but the reason for it needs to be written carefully at the end of the background section, about 1-2 paragraphs, considering the valid references and texts.
2. Theme and categories were not well-formed. The theme should be abstract and the items mentioned are not abstract.
The revisions of Table 2 require more consideration. The themes are not abstract. And should be changed to subheadings and revised throughout the text. Any numbering should be removed from the table text.
Category should be used instead of theme, subcategory should be used instead of category, and code should be used instead of semantic unit.
Author Response
- The authors have responded in the letter, but the reason for it needs to be written carefully at the end of the background section, about 1-2 paragraphs, considering the valid references and texts.
Thank you for your comments and suggestions. The reason has been included in the introduction section.
- Theme and categories were not well-formed. The theme should be abstract and the items mentioned are not abstract.
Thank you for your recommendations. The name of the categories and subcategories has been changed.
The revisions of Table 2 require more consideration. The themes are not abstract. And should be changed to subheadings and revised throughout the text. Any numbering should be removed from the table text.
Thank you for your suggestions. The table 2 has been revised, subheadings have been changed and numbers have been removed.
Category should be used instead of theme, subcategory should be used instead of category, and code should be used instead of semantic unit.
Thank you for your suggestions. The names have been changed: category by theme, subcategory by category and code by semantic unit.